# Assessing Physical Therapy Knowledge amongst the New Graduates in Saudi Arabia: Competency Examination across the Nation

**DOI:** 10.3390/healthcare10030579

**Published:** 2022-03-20

**Authors:** Fahad Alanazi, Muhammad Alrwaily

**Affiliations:** 1Department of Physical Therapy, College of Applied Medical Sciences, Jouf University, Sakaka 72388, Saudi Arabia; 2Division of Physical Therapy, School of Medicine, West Virginia University, Morgantown, WV 26506, USA; muhammad.alrwaily@hsc.wvu.edu

**Keywords:** assessment, competency examination, Saudi Arabia, NPTE, physical therapy

## Abstract

Given the increasing demand for more trained physical therapists in providing care to patients in Saudi Arabia, it has become vital to adequately assess individual physical therapy graduate academic learning and performance. Therefore, the present study aimed at evaluating the knowledge and skills of Saudi PT graduates. A competency examination adapted from a practice test that is commercially available and simulates the National Physical Therapy Examination (NPTE) was conducted. Out of 398 Saudi physical therapists that were approached with the examination link, 149 PT graduates consented to the study. Seventy questions were randomly selected by two individuals familiar with the content of PT programs in the United States and Saudi Arabia. The content outline of the selected questions followed the NPTE recommendations for body systems and non-systems. Each question was multiple choice with four answers. The examination was distributed electronically. Each participant was given 90 min to complete the examination. The passing score was set at 55%. Out of 149, only 6six (4.02%) participants passed the examination with an average passing score of 67% (range: 56–96%). In the primary domains of body systems, the score was highest in the endocrine domain (55.1%), followed by the integumentary (42.18%) and the neurology (40.9%) domains. In the non-system, participants had the highest score in the professionalism domain (89.8%). The highest mean knowledge score was obtained in the field of assessment (38.57%). PT graduates from Saudi Arabia performed poorly in the examination, demonstrating weak domain knowledge and skills.

## 1. Introduction

Physical therapy (PT) is one of the internationally recognized allied health professions practiced by qualified and trained or by licensed physical therapists as required by the national legislation of different countries [1]. PT is a dynamic profession that helps in treating acute or chronic pain, restoring movement, preventing physical impairments and disabilities resulting from trauma, injury, or illness in individuals [2]. Over the years, PT as a profession has evolved considerably, extending to different clinical fields of specialization, as well as education and research [1]. PT forms an essential part of the multidisciplinary healthcare team and aids in patient recovery and reducing patient hospital stay through physical examination, diagnosis, prognosis, patient education, physical intervention, rehabilitation, disease prevention, and health promotion [1].

PT education in the United States culminates with the National Physical Therapy Examination (NPTE) for licensure [3]. The NPTE is put in place to ensure that a PT student, before entering the workforce, achieved entry-level competence to practice safely and effectively [3]. The contents of NPTE assess the student’s acquisition of PT knowledge in examining and treating various body systems, including orthopedic, neuromuscular, cardiopulmonary, integumentary, endocrine, metabolic, gastrointestinal, genitourinary, lymphatic, and interaction between these systems [3]. The content of NPTE also extends to other aspects of PT such as equipment, technologies, therapeutic modalities, safety, professionalism, and research. A passing score on the NPTE is mandatory for entering practice [3].

Despite that PT education in the United States is standardized to ensure that students are taught by qualified faculty and have resources needed to support the curriculum, not all students pass the NPTE [4,5]. Several factors have been linked to poor performance in the NPTE, including student-level and institutional factors [4]. Student-level factors include academic performance, demographics, and scores on standardized aptitude tests. Institutional factors include class size, instructional approach, and research emphasis [4,5,6]. Collectively, the literature suggests that careful selection of students for admission to PT programs and continual improvement of quality of PT education are essential to ensure the competence of physical therapists before they practice.

In Saudi Arabia, PT is one of the predominant patient care professions, with the majority of physical therapists constituting the rehabilitation staff [7,8]. The number of universities offering PT education and students enrolling in PT programs in Saudi Arabia has increased from 6 to 14 over the past decade [8]. This increase in PT programs, however, has not been standardized to ensure the quality of education, and there is much variability in the PT programs offered in Saudi Arabia [8]. Some universities offer bachelor PT degrees, while others offer doctoral PT programs [7,8]. Also, there is variability among curricula and length of study from one university to another [7,8]. Graduates of PT programs in Saudi Arabia are licensed to practice upon successful completion of the internship year [9]. Physical therapists are only tested for competency if they were interrupted from clinical practicing for two years or more, or if they received their PT degrees from outside of Saudi Arabia [9]. In such situations, the Saudi Commission for Health Specialties (SCFHS) conducts a PT exam of 70 questions that an examinee must pass with a score of 55% [9].

Very few studies assessed the competency of newly graduated PT students in Saudi Arabia. Bindawas et al. used the Clinical Internship Evaluation Tool to assess the competency of intern students in their final year and showed that students had below-average scores (<3/5) in the examination, evaluation, differential diagnosis, and intervention skills [10]. Alshehri et al. surveyed physical therapists in Saudi Arabia in regards to their knowledge of evidence-based practice and showed that 70.2% had no formal training in their universities [11]. Alodaibi et al. used the Revised Neurophysiology of Pain Questionnaire (12 items) to assess students’ knowledge of pain neuroscience and showed that PT students in Saudi Arabia had less than optimal knowledge of pain neuroscience (mean score = 6.2/12) [12]. While these studies highlight deficits in certain areas of PT education in Saudi Arabia, there is a need for an overall assessment of knowledge that covers all aspects of PT education.

Despite the increased number of PT graduates in Saudi Arabia, their competency in various clinical skills has not been elucidated previously. Therefore, the purpose of this study is to assess the knowledge of Saudi PT graduates in the different body system and non-system domains of PT, patient examination, evaluation, and intervention.

## 2. Materials and Methods

### 2.1. Study Design

This study was a cross-sectional study approved by the Office of Human Research Protections at West Virginia University, United States (#1811364259).

### 2.2. Settings

The study was conducted using a single-session online test from 4 March 2019 to 30 April 2019.

### 2.3. Participants

To select the participants, an electronic link to the test was sent to email lists of specialized groups of physical therapists. Also, invitations to take the test were posted on several social media channels. Additionally, PT programs and hospitals in the northern, southern, central, and western regions of Saudi Arabia were invited to participate via phone or direct visits. A reminder to take the test was sent six times over the period of the study.

For inclusion, participants had to be new graduates of a PT program in Saudi Arabia within the past six months or be in their internship year. Participants who have PT degrees from outside Saudi Arabia were excluded.

### 2.4. Material

A competency examination for physical therapists in Saudi Arabia was conducted as a part of this research study intended to assess their knowledge of clinical applications, concepts and principles, and the minimum set of skills that are required for practicing safe and effective PT. The examination was adapted from two prominent study guides of the NPTE in the United States [13,14]. The study guides were intended to simulate the NPTE. The content outline of the selected questions followed the NPTE recommendations for (I) primary body systems such as cardiovascular and pulmonary, orthopedic, neuromuscular, integumentary, metabolic and endocrine, gastrointestinal, genitourinary, lymphatic, and system interactions having a range of items in different categories such as screening, assessment and interventions; and (II) non-systems such as equipment, devices, and technologies, therapeutic modalities, safety and protection, professional responsibilities, research and evidence-based practice [3]. 

Seventy questions related to the PT profession were randomly selected by two individuals familiar with the content of PT programs in the United States and Saudi Arabia. Each question had four multiple-choice answers, and participants could choose only one correct answer. Each participant was given 90 min to complete the examination. The passing score was set at 55%.

We chose 70 questions for two reasons. First, the SCFHS had 70 questions for PT licensure for those who obtained their PT degrees from outside Saudi Arabia; thus, we wanted participants to have a similar exam-taking burden [9]. Second, we believed that using 70 questions, as opposed to using 250 questions like the NPTE, was more feasible for our study to encourage participation without a huge time commitment.

We chose a 90-min examination time to match the time allocated to each question in the NPTE. In the NPTE, there were 250 questions that were completed in 5 h. This allocated 1.2 min to each question. Since we were using 70 questions, we allocated 1.2 min to each question. This adds up to approximately 90 min.

### 2.5. Data Collection

The test was conducted online to reach as many participants as possible in different provinces in Saudi Arabia, thus improving the representation of the sample. The test questions were entered, and data were collected using an electronic examination platform called ‘onlineexambuilder.com’. This platform had many test-taking features necessary for our study. It had a front page where the participant could be informed and consented to take the test as part of this research study and allowed for requesting participants’ information. It also had other tools such as restricting the number of attempts to one per participant, knowing how much time each participant spent taking the test, and exporting the data into a spreadsheet for further analysis.

After the participants consented, they were asked for demographics; then, they were presented with the test questions with a time bar showing progress. Because PT education in Saudi Arabia is in English, the test was in English. The test was not proctored, but we took several steps to reduce the bias of the participants’ ability to seek external help. Before the start of the exam, each participant was presented with a message that stated they should take the test independently without any form of help. We also presented questions and answers in random order. Additionally, the time allocated for each question (1.2 min/question) limited the chance that a participant leaves the test session to seek external help. Lastly, as soon as the participant finished and submitted the test, they were not allowed to take it again. The results of the test were automatically saved in the platform, which was protected with a password for subsequent analysis.

### 2.6. Sample Size

The number of new PT graduates is estimated to be 800 each year. Using calculator.net, we calculated the needed representative sample with a margin of error of ±10 and a 95% confidence interval to be 86 participants.

### 2.7. Statistical Analysis

Data were analyzed using SPSS 26 (Armonk, NY, USA). Descriptive statistics were used with sample demographics and characteristics.

## 3. Results

### 3.1. Participants’ Demographics and Specialty Characteristics

Out of a total of 398 physical therapists that were approached with the test link, only 149 responded and took the competency examination. Table 1 provides the demographics and examination scores by different characteristics. Only six (4.02%) participants passed the examination with an average passing score of 67% (range: 56–96%). 

### 3.2. Participants’ Score by Knowledge of the Primary Domains of the Body System 

The participants had the highest mean knowledge score of the endocrine system (55.1%), followed by the integumentary system (42.18%) and neurology (40.9%). The participants showed the least knowledge score of the system interaction domain (26.19%) (Figure 1).

### 3.3. Participants’ Score by Knowledge of the Non-System Domain

The participants showed the highest mean score or knowledge in the professionalism domain (89.8%), while the lowest score was in therapeutic modalities (17.69%) (Figure 2).

### 3.4. Scores of Participants Based on Different Categories of Patient Management

The participants obtained the highest mean knowledge score in the field of assessment (38.57%), followed by screening (36.95%), and lowest in the intervention (34.79%) (Figure 3).

### 3.5. Scores of Participants Based on Different Universities in Saudi Arabia

The highest mean score in the exam was obtained by the Saudi participants from Princess Nourah University (*n* = 7; 42.6%), followed by those from King Saud (*n* = 11; 38.5%) and Taibah University (*n* = 7; 38.3%), respectively. On the contrary, the participants from Hail university demonstrated the lowest mean score (*n* = 5; 26%) (Figure 4).

## 4. Discussion

Success in licensure examination plays a crucial role in a PT students’ professional career, allowing them to practice in a particular country. In Saudi Arabia, new graduates of the local PT program are not required to appear for licensing examinations [9]. Instead, the mere completion of a mandatory one-year clinical internship is considered enough for registration and practicing according to the codes of SCFHS, the licensing authority in the country [9]. The lack of licensing examinations for new PT graduates leaves their competency to practice safely and effectively uncertain.

The present study was undertaken to evaluate the knowledge and skills of new graduate physical therapists using a competency examination that resembles the outline of the NPTE [13,14]. The findings of the study showed that the vast majority of physical therapists who undertook the examination had poor performance, with an overall average score of 34% (Table 1). This low performance is similar to that found with previous studies that attempted to assess certain aspects of PT education. Bindawas et al. used the Clinical Internship Evaluation Tool to assess PT students’ management skills of the patient and found that both academic and clinical faculty rate the overall competency below average in various clinical skills [10]. Alshehri et al. found that 70.2% of physical therapists had no formal training in evidence-based practice in their universities or any other authorized training center [11]. Alodaibi et al. showed that PT students lacked sufficient pain neurophysiology education throughout the curriculum of PT education [12]. These studies, coupled with our findings, are disturbing and should call for an overhaul of PT education and licensure to practice in Saudi Arabia.

The present study also attempted to describe the students’ performance on several body systems (Figure 1). Of note, the endocrine system showed the highest score (55.1%), and system interaction had the lowest score (26.19%). However, orthopedic, neurology, and cardiopulmonary systems made up most of the examination questions since their contents constitute a large part of PT curricula. The scores of questions related to these systems were below average, suggesting that the examinee lacked knowledge of these systems. These findings suggest that PT curricula need to place more focus on systems relevant to patients commonly seen in clinical practices. 

Additionally, the present study attempted to describe the examinees’ performance on non-body systems (2). Examinees appear to have adequate knowledge in the professionalism aspect, which might be related to receiving proper instructions prior to entering the internship year or receiving adequate mentor guidance and facility orientation during the internship year. Examinees appear to have below-average knowledge of research, which is consistent with a previous study indicating that evidence-based practice education is lacking in PT programs in Saudi Arabia [11]. It was somewhat surprising that equipment and modalities had lower scores, as many PT curricula appear to place some focus on these forms of therapy, and practicing PT appears to be using these modalities frequently [15].

Moreover, the present study included questions related to clinical skills of screening, assessment, and intervention. Questions related to these clinical skills tended to examine the examinees’ ability of analytical thinking in analyzing case scenarios. Since the knowledge of these clinical skills appeared below average, it might be that PT programs should involve problem-based or case-based learning material throughout their curricula.

Further, the present study attempted to describe different performances of examinees across universities. Higher scores were achieved by the PT examinees from Princess Nourah, while the lowest were those from Jazan and Hail universities in Saudi Arabia. Several reasons may explain the differences in the performance of examinees in these universities. Princess Nourah is located in the urban capital of Saudi Arabia, Riyadh, while Jazan and Hail universities are located in rural areas. It might be related to differences in curricula from one university to another. For example, Princess Nourah offers a doctor of PT program that is restricted to female students, while other universities offer bachelor’s degrees in PT. It might be that PT students in urban areas of Saudi Arabia have access to training more readily than those in rural areas. The differences in examinees’ performances across universities call for more standardization of PT education in Saudi Arabia to ensure rigorous graduation standards. An example of such standardization is similar to that of the Commission on Accreditation in Physical Therapy Education in the US [16].

Our study did not attempt to examine factors related to examinees’ success in the examination; however, various studies have identified several predicting factors in determining success in the licensure examinations [4,5,6]. These factors include research-intensive PT programs, increased laboratory contact hours, institutional status (public versus private), increased funding, lower student-to-faculty ratios, class size, and rigorous graduation standards [4,5,6]. Furthermore, examinees’ academic difficulty during the training in PT schools and the first- and third-year grade point average have also been seen as a significant independent predictor of success in licensure examination [4,5,6,17,18]. We are unaware of studies that explored factors related to PT graduates’ academic performance in Saudi Arabia. 

In Saudi Arabia, the bachelor’s degree in PT is a 5–6 years program with first-year courses focusing on Islamic studies, languages, and basic sciences [8]. Second-year courses are introductory PT procedures, and third- and fourth-year courses emphasize clinical coursework in neurology, internal medicine, radiology, psychology, orthopedics, and geriatrics. Fifth-year is a one-year internship in clinical PT practice with a PT research project designed with the objective to increase and combine knowledge and skills in different PT areas such as evaluation, clinical examination, diagnosis, prognosis, and intervention [7,8]. Post-graduation program focusing more on research is provided at some universities like King Saud University. Doctoral programs in PT are not widely available in Saudi Arabia [8]. 

Achieving success with good passing scores in licensure or assessment examinations is of great importance and may also have significant consequences for several stakeholders such as PT students themselves, the reputation of PT programs, patients, PT institutes including prospective faculty and new applicants who may compare different institutions’ pass rates as a measure of the merit of the individual institution [19].

## 5. Limitations

This was a convenience sample, and so we could not exclude the possibility of selection bias. Even though we attempt to reach as many PT graduates as possible, we cannot be sure that our sample is representative of the entire PT practitioner population. Our power calculation included a margin of error of ±10, and we cannot exclude the possibility of sampling bias. Our study included volunteers who may have chosen to appear for the test with a desire for professional self-development and personal growth in practice. While the examination we used is taken from a popular study guide, it still lacks validation as a measurement of examinees’ knowledge. The examinees’ were not proctored during the examination, so we cannot be sure that they did not seek external help. Before undertaking the licensure examination, graduates typically engage in an extensive review of the material. However, the participating examinees did not have the chance to study for such an exam. Our study was cross-sectional, so it may not reflect the performance of students graduating in different semesters. 

## 6. Future Directions

Because of the ever-changing health care environment, it becomes vital that formal assessments of physical therapists’ competency skills must be performed [20]. Regular modifications and improvements in the assessment tools should also be brought up according to the practice settings [21]. This strategy would not only address the needs of various PT practice settings for different patients but would also bring about necessary changes at the institutional and national levels in terms of educational quality, academic policy and practice, accreditation, licensure policies, and other similar issues. Updating the curriculum and introducing more doctoral programs in PT education taught by experienced faculty members would further ensure that quality students graduate from the institutes. Additionally, the 95 elements, categorized under the specific knowledge, skills, attitudes, and professional behaviors as identified in a Delphi study, are recommended to be used as guidelines when determining PT graduates’ readiness for a first full-time clinical experience [22]. 

## 7. Conclusions

Our findings demonstrated low levels of knowledge among PT graduates in Saudi Arabia. As the number of students enrolling in PT programs is increasing in Saudi Arabia, it is imperative to ensure high quality, standardized PT education and clinically-oriented and research-based courses in all universities across the nation. The establishment of a strong professional PT council may further help provide guidance in this respect.

## Figures and Tables

**Figure 1 healthcare-10-00579-f001:**
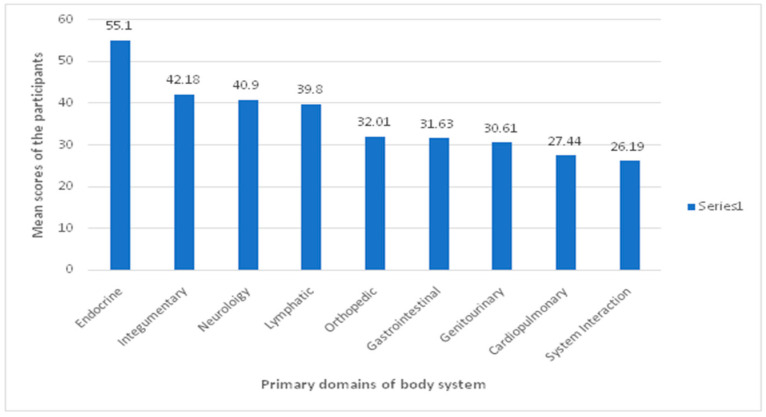
Mean knowledge score of participants in different domains of the body system.

**Figure 2 healthcare-10-00579-f002:**
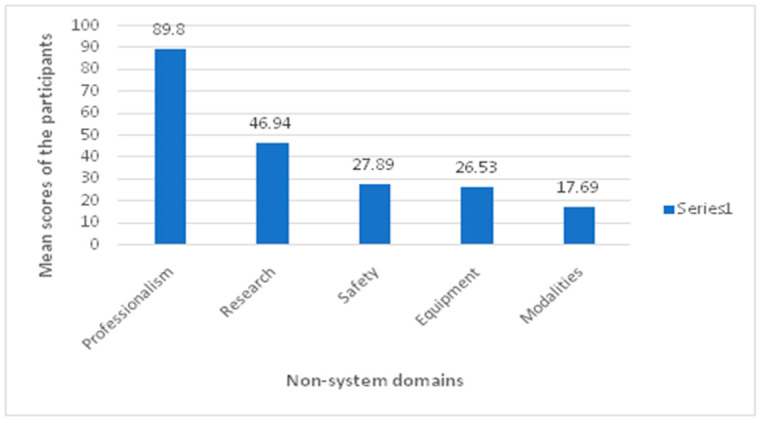
Mean knowledge score of participants in different non-system domains.

**Figure 3 healthcare-10-00579-f003:**
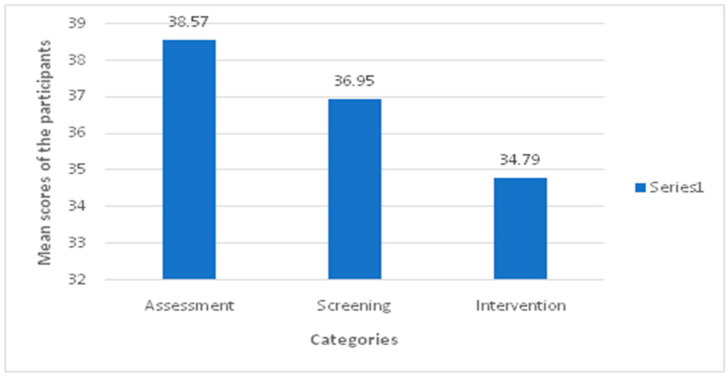
Mean knowledge score of participants in different categories.

**Figure 4 healthcare-10-00579-f004:**
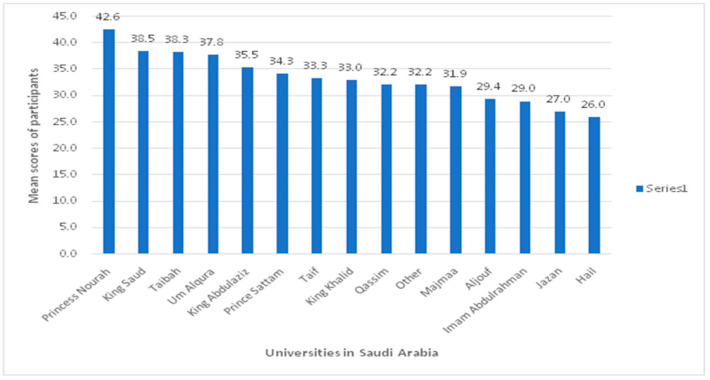
Mean scores of participants from different universities in Saudi Arabia.

**Table 1 healthcare-10-00579-t001:** Participants’ demographics and examination scores by different characteristics.

Groups	Number of Participants	Average Scores
Gender		
Males	61	32%
Females	88	34%
PT Level		
Final Internship Year	47	36%
Fresh PT Graduate (within 6 months)	102	32%
Overall Score	149	34%

## Data Availability

Not applicable.

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
