# Peer review of "Assessing Physical Therapy Knowledge amongst the New Graduates in Saudi Arabia: Competency Examination across the Nation"

_healthcare, 2022, doi:10.3390/healthcare10030579_

Round 1
Reviewer 1 Report
Thank you for the opportunity to review this manuscript. The researchers present a novel idea to assess the knowledge base of physical therapists practicing in Saudi Arabia. However, there are methodological flaws and significant errors with the results section (missing Table 1 that the authors reference) and errors with the citations in this manuscript. There are missing as well as incorrect citations. The test questions from the national physical therapy examination (NPTE) are highly protected. The authors used examination questions from a NPTE preparation course, but it is unclear if they had permission to use those questions and modify the test questions for this study. If that is the case, then the authors need to clearly state that. The limitations section must be greatly expanded upon.
Introduction
The authors would benefit from a more in-depth literature review to fully understand and explain what the national physical therapy examination (NPTE) measures and how this translates to safe and effective patient care in the United States.
Page 1, Line 35 – “chronic pain” – not plural
Page 1, Lines 38-39 – orthopedic and musculoskeletal are redundant in the same sentence
Page 1, Line 40 – the authors need a better reference to understand and describe the domains of practice
Page 2, Line 57 – Reference 7 is missing from the reference list
Line 52-55 – The authors assert that an NPTE outcome is linked to high quality patient care and need to cite this statement
Methods:
Need additional information about the demographics about the population of physical therapists practicing in Saudi Arabia.
Materials:
Line 74-75 - The test questions from the national physical therapy examination (NPTE) are highly protected. The authors used examination questions from a NPTE preparation courses, but it is unclear if they had permission to use those questions and modify the test questions for this study. If that is the case, then the authors need to clearly state that.
Line 87 – Why was the passing score set at 55% for this study? Elaborate.
Line 105 – The authors state that the test was in English. Why is this? What is the primary language in Saudi Arabia? Did the participants speak English as a first or additional language? Did the authors really test physical therapy knowledge or did they test English? This needs to be considered in the limitations section.
The authors appeared to set up the test to mimic the NPTE But the test was only 70 questions and participants had 90 minutes to complete the test. How was it determined that this amount of time was sufficient for answering 70 questions? Neither of these two requirements are consistent with the NPTE which has 250 items.
Were the test questions tested ahead of time with a pilot group? Did the authors receive feedback about the test questions they selected?
Data collection
The test was not proctored. Was it truly a test of knowledge if it was potentially an open-book test?
Results
Line 119. The authors reference Table 1. There are no tables in this manuscript.
Discussion
Line 163-164 – The authors state that the outline of the exam used in this study was based on the NPTE. However, the sections that the authors describe “Endocrine” and “Professionalism” are not consistent with the NPTE Content Outline effective January 2018. I believe that the authors are referencing “Metabolic and Endocrine” and “Professional Responsibilities”
Line 167 – The authors state that a finding was “significantly low.” Without inferential statistics, the authors cannot make any assertations about statistical significance.
Line 169 – this is new information and not stated in the results section
Line 171-174 – statement about internship training requires a citation
Line 174-176 – information about interns is also new information which was not introduced in the results section. The authors make an assertion that they cannot prove through the results of this study.
Line 185-186 – the authors assert that professionalism cannot be learned through theoretical courses – need a citation for this statement
Line 190 – The Guide to Physical Therapy Practice – needs to be capitalized
Line 198-199 – “such skills” – it is unclear what the authors are referring to
Line 202-205 – Elaborate or delete
Line 226-227 – the reference for this statement is inaccurate
Line 229 – Reference 21 is not accurate for this statement
Line 242-245 – Needs a citation
Please expand on the Limitations section. There are far more than stated by the authors.
Author Response
Thank you for your comments that we find very helpful in improving our manuscript.

Reviewer 2 Report
This educational research provides an insight toward the knowledge of new graduates PTs in KSA. Please find my following comments:
- The title should be more related to study contexts as: palliative care, and body systems
- The introduction should provide more details about the body of the current evidence, limitation, rational, and similar studies.
- I strongly recommend to report your study according to STROBE guidelines
- Is the NPTE a valid tool to assess the knowledge for new graduates? Please explain that
- I strongly recommend to report about the sampling, and power sample size calculation
- Why the authors did not analysis statically (as t-test) the knowledge of students according to the students characteristics such as age, gender, university of graduation, and highest academic certificates,
- I would like to see more clear educational and research recommendations in the discussion
Author Response

(The authors gave the same response as above.)

Round 2
Reviewer 1 Report
Comments:
Thank you for the opportunity to review this revised manuscript. While the authors have managed to improve the Introduction section, they have done so often without citations, and this must be corrected. Otherwise, this manuscript is based on anecdotal evidence. Without appropriate citations the authors are unable to contribute to the existing body of knowledge. Additionally, reference to the NPTE must be removed from the abstract as the examination used in this study is NOT based on the NPTE. Instead, it is based on an NPTE practice test. I remain concerned that the authors have not obtained permission to modify the practice test they used in this study. While the authors included Table 1 which was missing from the original manuscript, they now reference a Table 2 in the discussion section. There is no Table 2 in this manuscript. The authors also reference Appendix 1 but there is no appendix in this manuscript. T
Abstract:
Line 11 “palliative care” – the authors should remove the word palliative as physical therapists provide extensive care beyond caring for patients with serious health conditions such as cancer or heart failure.
Line 14 Inaccurate and need to correct. The exam in this study was NOT based on the NPTE but from commercially available practice tests.
Line 16-17 Should be changed to “149 eligible physical therapists consented into the study”
Introduction
Line 41 – delete “certain activities”
Line 45 needs a citation
Line 47 delete “typically”
Line 48 Licensure does not need to be capitalized
Line 49 “entry-level” spelled incorrectly
Lines 47-56 have no citation. How do the authors know this?
Line 57-59 Missing a citation about pass rate on the NPTE.
Lines 59-65 missing citation
Line 119-120 “To select participants, an electronic link to the test was sent to email lists of specialized groups of physical therapy”. I Believe that the authors mean “physical therapists”
Line 123 What does the abbreviation MSK stand for?
Table 1 – How have the authors operationally defined “Experienced”
Line 142 – the authors reference Appendix 1 but there is no appendix in this manuscript
Discussion
Line 258: The authors reference Table 2. There is no table 2 in the results section.
Line 271 “preset study” I believe the authors mean “present study”
Line 273 “Orthopedic” should not be capitalized
Line 283-285 missing citation
Line 291 and line 341 “examin” spelled incorrectly
Author Response
Dear editor and reviewers,
Thank you very much for helping us improve our manuscript further. Please find below our responses to your comments.
Reviewer 1
Comment
Thank you for the opportunity to review this revised manuscript. While the authors have managed to improve the Introduction section, they have done so often without citations, and this must be corrected. Otherwise, this manuscript is based on anecdotal evidence. Without appropriate citations the authors are unable to contribute to the existing body of knowledge. Additionally, reference to the NPTE must be removed from the abstract as the examination used in this study is NOT based on the NPTE. Instead, it is based on an NPTE practice test. I remain concerned that the authors have not obtained permission to modify the practice test they used in this study. While the authors included Table 1 which was missing from the original manuscript, they now reference a Table 2 in the discussion section. There is no Table 2 in this manuscript. The authors also reference Appendix 1 but there is no appendix in this manuscript. T
Response
Thank you for your time for helping us improve our manuscript. We have added references now to the introduction section. We also removed reference of the NPTE from the abstract. We understand the concern about obtaining permission, however, we believe that we are under “fair use” doctrine that allows limited use of copyrighted material for research and scholarship. We removed the word “Table 2”. We added Appendix 1.
Comment:
Abstract:
Line 11 “palliative care” – the authors should remove the word palliative as physical therapists provide extensive care beyond caring for patients with serious health conditions such as cancer or heart failure.
Response
“Palliative care” has been removed
Comment
Line 14 Inaccurate and need to correct. The exam in this study was NOT based on the NPTE but from commercially available practice tests.
Response
We changed the sentence to:
“A competency examination adapted from a practice test that is commercially available and simulate the National Physical Therapy Examination (NPTE) was conducted.”
Comment
Line 16-17 Should be changed to “149 eligible physical therapists consented into the study”
Response
We changed the sentence:
“149 PT graduates consented into the study”
Comment
Introduction
Line 41 – delete “certain activities”
Response
“certain activities” deleted
Comment
Line 45 needs a citation
Response
Citation added
Comment
Line 47 delete “typically”
Response
“typically” deleted
Comment
Line 48 Licensure does not need to be capitalized
Response
“Licensure” was corrected to “licensure”
Comment
Line 49 “entry-level” spelled incorrectly
Response
“entry-level” corrected
Comment
Lines 47-56 have no citation. How do the authors know this?
Response
Citation was added
Comment
Line 57-59 Missing a citation about pass rate on the NPTE.
Response
Citations were added
Comment
Lines 59-65 missing citation
Response
Citations were added
Comment
Line 119-120 “To select participants, an electronic link to the test was sent to email lists of specialized groups of physical therapy”. I Believe that the authors mean “physical therapists”
Response
“physical therapy was corrected to “physical therapists”
Comment
Line 123 What does the abbreviation MSK stand for?
Response
MSK was deleted
Comment
Table 1 – How have the authors operationally defined “Experienced”
Response
This was a mistake in the table. We have now included the correct table.
Comment
Line 142 – the authors reference Appendix 1 but there is no appendix in this manuscript
Response
Appendix 1 has been added
Comment
Discussion
Line 258: The authors reference Table 2. There is no table 2 in the results section.
Response
“Table 2” was removed. It was a mistake.
Comment
Line 271 “preset study” I believe the authors mean “present study”
Response
“Preset study” was corrected to “present study”
Comment
Line 273 “Orthopedic” should not be capitalized
Response
“Orthopedic” was corrected to “orthopedic”
Comment
Line 283-285 missing citation
Response
Citations were added
Comment
Line 291 and line 341 “examin” spelled incorrectly
Response
“examin” was corrected to “examine”
Reviewer 2 Report
Thank you for revising the manuscript. Please find my follow minor comments:
- The new paragraphs and sentences need supportive references.
- You mentioned that a margin of error of ±10. Please consider that as a limitation. There is a high risk of sampling bias.
-
Author Response
Dear editor and reviewers,
Thank you very much for helping us improve our manuscript further. Please find below our responses to your comments.
Reviewer 2
Comment
Thank you for revising the manuscript. Please find my follow minor comments
Response
Thank you very much.
Comment
- The new paragraphs and sentences need supportive references.
Response
Supporting references were added
Comment
- You mentioned that a margin of error of ±10. Please consider that as a limitation. There is a high risk of sampling bias
Response
We added a sentence: “Also, our power calculation included margin of error of ±10, and we cannot exclude the possibility of sampling bias”.